# Two-Stage Penile Reconstruction after Paraffin Injection: A Case Report and a Systematic Review of the Literature

**DOI:** 10.3390/jcm12072604

**Published:** 2023-03-30

**Authors:** Luigi Napolitano, Claudio Marino, Angelo Di Giovanni, Assunta Zimarra, Alessandro Giordano, Carlo D’Alterio, Gianluigi Califano, Massimiliano Creta, Giuseppe Celentano, Roberto La Rocca, Claudia Mirone, Gianluca Spena, Alessandro Palmieri, Nicola Longo, Ciro Imbimbo, Marco Capece

**Affiliations:** 1Department of Neurosciences, Reproductive Sciences and Odontostomatology, University of Naples Federico II, 80100 Naples, Italy; 2Multidisciplinary Department of Medical, Surgical and Dental Sciences, University of Campania Luigi Vanvitelli, 80100 Naples, Italy

**Keywords:** penile injection, penile size, penile paraffinoma, two-stage penile reconstruction, paraffin

## Abstract

Background: Penile injection of foreign materials is an obsolete practice often performed by non-medical personnel in order to enlarge penile size. Methods: A systematic review of the literature from 1956 to 2022 was conducted in accordance with the general guidelines recommended by the Primary Reporting Items for Systematic Reviews and Meta-analyses (PRISMA) statement. We included full papers published from 1956 to 2022. We also described a case report of a 23 year old Bulgarian male affected by penile paraffinoma who underwent a 2-stages surgical technique. Results: A total of 152 cases have been reported, with a median age of 37.9 ranging from 18 to 64 years. Six different techniques have been described in the whole literature: bilateral scrotal flap, simple excision of the paraffinoma with primary closure, two-stage scrotum skin flap, medial prepuce-soprapubic advancement flap technique and penile reconstruction using split thickness skin graft (STSG) or full thickness skin graft (FTSG). An analysis of the distribution among early and late complications was then carried out. Conclusion: In our experience, among the variety of surgical techniques described, a two-stage penile reconstruction using scrotal skin results in excellent cosmetic and functional outcomes, with a low rate of complications.

## 1. Introduction

Paraffin penile injection is an old, obsolete practice for penile girth augmentation. Paraffin is injected to increase penile girth or length or to a perceived sense of sexual pleasure [1]. Foreign bodies or materials have been injected into the human body since ancient times, but oil injection was first described in 1899 by Robert Gersuny as testicle substitution in patients who had received a bilateral orchiectomy because of tuberculous epididymitis [2,3].The term paraffinoma was introduced by Newcomer and Grahamin in 1971 to describe abnormal histopathologic findings after the injection of foreign substance containing straight-chain hydrocarbons, such as paraffin, vaseline, silicone or mineral oil [4]. Despite the well-known potential devastating complications, this practice is still very common among Asian and Eastern European populations [5]. In fact, despite the initial good results, several short- and long-term destructive complications have been described.

Indeed, the injection of any foreign material into the body leads to the development of an inflammatory and fibrotic process, called foreign body reaction (FBR) [6]. Immune system plays a pivotal role in this inflammatory reaction with an acute and chronic phase [7]. The acute phase is characterized by local edema, redness, tenderness, pain and a cascade of cellular events immediately after implantation [7]. In the first time, neutrophils surround the implant and begin to release factors which promote the progression of the inflammatory process. These factors provide recruitment of other acute inflammatory cells as well as macrophages, histiocytes, lymphocytes and giant cells [8]. Monocytes, once arrived, begin to differentiate into macrophages which through production of several pro-inflammatory products as well as TNFα, tumor necrosis factor α, and interleukins IL-1b, IL-6, and IL-8, self-sustains own recruitment and proliferation. Macrophages adhere to and cover the surface of the implant, forming a defined and isolated space [9]. Over a period of weeks to months this inflammatory process develops into a chronic fibrotic response. Macrophages population play a fundamental role in the formation of a fibroblast and ECM-rich capsule that covers and isolates the implant. This process leads to granuloma formation, characterized by extracellular matrix proteins, myofibroblasts, and foreign body giant cells (FBGCs), derived from macrophages fusion [6].

Despite the well-known pathophysiology, this primitive technique is still practiced illegally by non-medical personnel, increasing the chances of having disastrous side effects [10]. Over the last few years, several cases of paraffin injection have been reported in the literature [11]. Even after many years patients who have undergone similar injections, can develop complications that require, in most of the cases, surgical intervention in order to remove the foreign body [12]. Nowadays, no standardized treatment protocols exist; non-surgical therapy, as well as topical cream or steroid injection, was found to be ineffective. Surgical removal of the foreign material and granuloma, followed by appropriate skin grafting is the standard of care. Different techniques have been reported according to patients’ characteristics and surgeons’ preferences. The purpose of treatment is to restore penile function and adequate cosmetic appearance [13]. Here we described a case of paraffinoma treated at our department using a two-stage reconstruction technique and performed a systematic review of the literature to provide an updated overview of evidence about treatment and outcomes of penile paraffinoma.

## 2. Case Report

A 23-year-old Bulgarian man presented to our hospital with penile pain and erectile dysfunction. He reported to have received liquid paraffin injection 2 years before in his country by non-medical personnel in order to increase his penile girth. He referred, after a short initial period of normal sexual intercourse and no complications, pain and difficult erections. On his physical examination the penis appeared swollen with an irregular and abnormal shape and a phimotic attitude (Figure 1). Irregular masses were palpable at the upper right part and at the base of the penis without involving scrotal tissues. The foreskin appeared thin and not easily dissociable from the underlying tissues. The preputial and penile shaft skin displayed a typical appearance of paraffinoma. No reactive lymph nodes were palpable. No voiding dysfunction was reported. No specific findings were obtained from a general blood test, liver function test or hepatitis test. A penile RMN was performed before undergoing surgical treatment. The magnetic resonance showed that all tissues were extremely and diffusely thickened. The maximum thickness (17 mm) was in the distal part with a heterogeneous signal in T1 and T2 (Figure 2). Inhomogeneity of the signal was less at the deepest levels. No involvement of corpora cavernosa, corpus spongiosum or glans, with spots of liponecrosis. The patient underwent surgical treatment. We decided to perform a two steps procedure in order to get the best aesthetical result. The first step lasted around 2 h and 30 min. After subglandular circumferential incision, the healthy skin of the penis was tangibly differentiated from the infiltrated skin. Excision of all the infiltrated tissue was performed (Figure 3). Penile shaft measurement was performed after inducing an artificial erection with sterile saline solution (Figure 4). Scrotal skin was extended in order to draw the necessary landmarks based on the measurements of the erected penis. Later, a subfascial scrotal tunnel was created and the penis was scrotalized (Figure 5) frame dal video. An aspiration drain (ch10) was placed. Layers closure. Semicompressive dressing was done. Placement of Foley catheter (ch16). Catheter was removed on the 5th post-op. The patient was dismissed on the 7th day post-op. No short-term complications were noticed. Two weeks later, at the follow up visit, the patients presented no pain, no adverse reactions, no edema. Histopathological examination of the removed tissues showed compatible characteristics with clinical diagnosis of paraffinoma. Epidermidis showed regular aspects. The patient was asked to massage the penis/scrotal skin with a moisturizing and emollient cream twice a day for the following months. After 6 months at the physical examination the scrotum resulted in more elasticity and tenderness. A second step of penile reconstruction was then performed. After surgical field preparation, necessary landmarks for the following scrotoplasty were measured. A scrotal access was performed up and not beyond the vaginal tunica and the largest possible scrotal tissue was earned in view of the reconstruction (Figure 6). Then ventral closure of the skin was done, wrapping the shaft of the penis and following closure of the scrotus with V-Y technique (Figure 7). Semi compressive medication was applied. Foley catheter (Ch 14) was placed. The final result at 6 months of follow up is reported in Figure 8.

## 3. Materials and Methods

A systematic review of the literature was then conducted in accordance with the general guidelines recommended by the Primary Reporting Items for Systematic Reviews and Meta-analyses (PRISMA) statement (Figure 9) [14].

The search was performed in the Medline (US National Library of Medicine, Bethesda, MD, USA), Scopus (Elsevier, Amsterdam, The Netherlands) databases, and Web of Science Core Collection (Thomson Reuters, Toronto, ON, Canada). The following terms were combined to capture relevant publications: penile paraffinoma, penis paraffinoma, paraffine injection. We included full papers published from 1956 to 2022 that met the following criteria: original research, full text, injection of liquid paraffin into the penis, case report, case series, human studies and surgical treatment described. Reference lists in relevant articles and reviews were screened for additional studies. Abstracts (with no subsequent full-text publications) and unpublished studies were not included such as articles written in non-english languages.

## 4. Literature Search

Two authors (C.M. and A.Z.) reviewed the records separately and individually to select relevant publications, with any discrepancies resolved by a third author (A.d.G.). To assess the eligibility for the systematic review, PICOS (participants, intervention, comparisons, outcomes, study type) criteria were used. PICOS criteria were set as follows: (P)articipants—Patients; >18 years old who had received penile injection of paraffina;(I)ntervention: surgical intervention in men undergo paraffin penile injection; (C)omparator:different surgical technique; (O)utcome: functional outcomes in terms of sexual activity, psychological outcomes, aestetic outomes. Study types: case control, case series, case report [15].

## 5. Data Collection

The following data were extracted from the studies included: first author, year of publication, median age of the patients, country, clinical presentation (at the first visit), symptoms, modality of injection, time of injection, surgical technique, operation time, results (in terms of aesthetic and functional outcomes), early and late complications.

The methodological quality of case reports and case series was performed according to Murad et al. [16]. (Table 1). The CARE checklist, according to care guidelines 2013, was also performed for each study included (Table 2).

The search strategy revealed 69 results. The screening of titles and abstracts revealed 63 full-text articles eligible for inclusion. Further assessment of eligibility, based on full-text papers, led to the exclusion of 16 papers. In the screening phase other 7 articles were excluded because the surgical technique was not described, 16 articles were excluded because the substance injected was not specified or was different from paraffine. Finally, other 2 studies were excluded from our review because the paraffine was not injected (topical application *n* = 1) and the injection was not on the penis (*n* = 1). In conclusion 22 papers involving 152 patients were included in the final analysis.

## 6. Results

A total of 152 cases have been reported, with a median age of 37.9 ranged from 18 to 64 years. Two papers did not report the age of the patients. In 28 cases (18.4%) patients underwent self-injections, while in 57 patients (37.5%) the injections were performed by non-medical personnel. In 67 patients (44.1%) this information was not available. Several clinical presentations have been reported: an irregular mass on the penis (*n* = 49, 32.2%), ulceration (*n* = 40, 26.3%), pain (*n* = 23, 15.1%), deformity (*n* = 21, 13.8%), granuloma (*n* = 20, 13.2%), necrosis (*n* = 14, 9.21%), skin changes (*n* = 10, 6.58%), phimosis (11, 7.24%), erectile dysfunction (*n* = 6, 3.95%), voiding difficulty (*n* = 4, 2.64%), oedema (*n* = 2, 1.32%), pruritus (*n* = 1, 0.66%), irregular mass on the scrotum (*n* = 1, 0.66%).

The median interval between injection and clinical presentation was 5 years (range: 5 days–35 years). All the patients underwent surgical intervention.

Six different techniques have been described in the whole literature.

The first and most used technique was the bilateral scrotal flap described in 11 studies and performed in 94 patients (61.8%) [4,18,21,23,24,26,27,28,32,33] Thirty-five out of 94 underwent a concomitant scrotoplasty [23,32].

Simple excision of the paraffinoma with primary closure has been described in 6 studies with a total of 42 patients (27.6%) [17,18,19,20,29,34]. Twelve patients in 5 studies underwent a split thickness skin graft (STSG) harvested from the upper thigh (7.9%), [12,18,22,30,35].

Two-stage scrotum skin flap was only described in 2 studies with a total of 2 patients involved (1.32%) [21,33]. Finally, a medial prepuce-soprapubic advancement flap technique was described in one patient (0.66%) as well as a full thickness skin graft (FTSG) in another one(0.66%) [31]. Histological findings were reported in 55 cases (36.18%): lipid-filled vacuoles with scleroted stroma and marked chronic inflammation.

Regarding the early and late complication, the following have been described: In Bilateral scrotal flap procedure early complications, which occur in less than 8 weeks, have been described in three studies: Wound disruption (4 patients, 2.63%,), oedema (three patients, 1.97%) penile rugae (one patient, 0.66%). Late complications, described in 56 patients (36.84%) were: Delayed wound healing (*n* = 20, 13.2%), infections (*n* = 17, 11.2%), Shortened penis (*n* = 4, 2.6%), scar contracture (*n* = 5, 3.3%), decreased scrotal size (*n* = 6, 3.94%), lower abdominal stretching (*n* = 2, 1.32%), horseshoe relapse on the base of the penis (*n* = 1, 0.66%).One patient developed necrosis about 0.4 cm (0.66%) while 2 patients developed wound dehiscence (13.3%) (Table 3). After a split thickness graft procedure, 5 patients developed oedema and a feeling of tension during erection, after a split thickness skin graft surgery. No late complications were described. Among patients who underwent a paraffinoma excision with primary close, one patient developed, as late complication, recurrent lesion (0.66%). No early complications were described after this procedure. No complications were described after Two-stage scrotum skin flap, medial prepuce-soprapubic advancement flap technique and full thickness skin graft surgery.

The analysis of the distribution of the complications showed that, among early complications, the most common was Oedema (44.4%) followed by feeling of tension during erection (27.8%), wound disruption (30.8%) and plication of the skin (5.6%) (Figure 10).

Instead, among late complications, the most common was delayed wound healing (33.9%) followed by infections (28.8%), decreased scrotal size (10.2%), scar contracture (8.5%), shortened penis (6.8%), lower abdominal stretching (3.4%), wound dehiscence (3.4%), recurrent lesions (1.7%), horseshoe relapse on the base of the penis (1.7%) and necrosis (1.7%). (Figure 11)

All the long time results described were positive with general full recovery with resumption of sexual activity, resolution of ED, better physical and psychological out-comes of the patients.

## 7. Discussion

Penile size has always been considered as a symbol of power [25]. Injection of foreign materials under the penile skin in order to make the penis bigger has been reported in several countries [36]. However, the injections of foreign materials are usually followed by complications, including penile deformity, skin necrosis, decreased erectile function and painful intercourse [17].

In our analysis the most frequent clinical presentation of paraffine injection was an irregular mass on the shaft of the penis (32.2%) followed by ulceration (26.3%) and pain (15.1%).

The injection of paraffin inside the body creates an immediate inflammatory reaction, followed by a relatively long latent phase resulting in the formation of multiple foreign body granulomas, known as paraffinomas [37]. When this substance is injected into the penis the formation of paraffinoma may cause severe damages, mainly due the fact that the penis does not have a fixed shape or position [38]. In most of the cases a surgical intervention is required, and the present paper aims to present a two-stage surgical technique accompanied by a systematic review of all surgical procedures described in literature.

To date, this is the first systematic review focusing on paraffin injection into the penis. No official guidelines have been published about the management of paraffinoma or its complications, and early intervention is recommended, with the aim of restoring penis function and reaching an acceptable cosmetic appearance.

In the present review many surgical techniques have been described: Scrotal Skin Flap, Cecil’s inlay operation, Split thickness Skin Graft (SSG) are the most commonly used options. One hundred fifty-two patients have been analyzed in our systematic review and more than half of them underwent a complete excision of the penile skin with a single stage bilateral scrotal skin flap. Clearly this is an acceptable option if you have a good compliance of the scrotum that allows an easy reconstruction of the penis without the use of skin graft. Penile resurfacing with scrotal skin is a relatively simple, effective, and reliable reconstructive procedure that could be performed in single- or double-stage. Jeong et al. reported good aesthetic and functional outcomes [23]. This technique can be an effective and reliable method especially for patients with foreign material that involves the Buck’s fascia. In fact, in such cases the use of a skin graft, either split or full, may be considered risky as it may not receive the proper blood supply from the tunica albuginea, which is less vascularized than dartos fascia.

Single excision of the granuloma via circumferential subcoronal incision with skin preservation, as reported in many published studies, can be performed when the mass is not extensive [18,29,39]. This option should be carefully indicated since there might be a risk of skin devascularization that will lead to skin necrosis and a subsequent unpredictable operation [40].

The two-stage technique has only been described in a few patients (2 studies) [21,33]. In our case report, we performed this technique, despite the vast majority of the authors prefer a single stage technique. The main reason for choosing this technique is the combination of a small size of the scrotum in a patient who does not want a skin graft as he did not accept the possible cosmetic appearance of a graft. At the first stage, the granuloma was completely excised followed by burying the penile shaft underneath scrotum skin. The following months the moisturizing cream and massages gave the skin of the scrotum the possibility of elasticizing and widening. At the second stage we performed the reconstructive procedure. Although this technique has the disadvantage of requiring longer times and a double intervention, it gives good aesthetic and functional outcomes, possibly better than what can be expected from the use of grafts. Thus, it may be considered a good option for young and compliant patients.

In fact, as described by Katinka et al. and Sallaudin et al. in 2019, we report no complications after two- stage scrotum skin flap procedure [18,27]. In our experience, we performed it in double-stage considering the extension of the scrotum at the moment of the diagnosis. The scrotal skin has the advantage of being quite elastic, and it tends to increase in dimensions when properly stretched. Indeed after 6 months of appropriate stretching and massaging from the patient it reached the number of tissues needed to perform a scrotoplasty without reducing the size of the scrotum and compressing the testicles [41]. In such cases three to six months of exercises could be enough to gain the proper dimensions. The patient had great satisfaction from the final result and good aesthetic and functional outcome. However, this data should be considered with caution: the major limitation of this study is related to the methodological quality of available data. Indeed, we are conscious that data retrieved from case reports are considered the lowest level of evidence. Despite this, they are considered to fill an important role as a data source for heterogeneous conditions as well as paraffin injection [21]. Although methodologically challenging and burdened with a high risk of bias, systematic reviews of case reports can provide a useful addition to evidence-based medicine and can provide the basis for hypothesis generation [42,43]. Unfortunately, the heterogeneity of these cases does not allow us to make adequate comparisons. Further it is necessary to improve the evidence level adopting in these patients by setting standardized reporting systems such as the CAse REport (CARE) checklist.

## 8. Conclusions

Nowadays men still look for penile size augmentation to increase their manhood. According to this behavior, men undergo procedures that may be a potential risk for adverse effects [2,44].

Instead of the expected, psychological, functional and esthetic outcomes, severe adverse effects can occur after the injection of any material into the penis, and complex surgical operations are needed in order to resolve such issues [4,45]. Although no standardized protocol exists, surgery remains the gold standard treatment to restore aesthetically and functionally the penis. According to the severity of the problem and surgeon preference, single-stage or multiple-stage reconstructions can be used. In our experience a two-stages penile reconstruction using scrotal skin results in excellent cosmetic and functional outcomes, with a low rate of complications. Despite this, there is a need for preventative measures through public health awareness and education campaigns in areas where this practice seems to be prevalent.

## Figures and Tables

**Figure 1 jcm-12-02604-f001:**
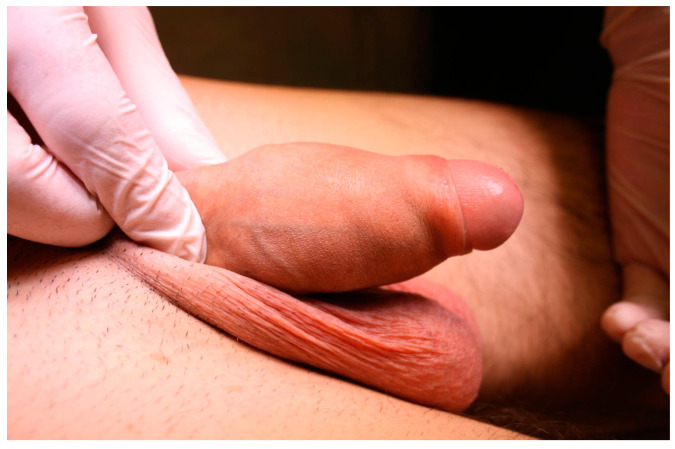
Swollen and phimotic penis.

**Figure 2 jcm-12-02604-f002:**
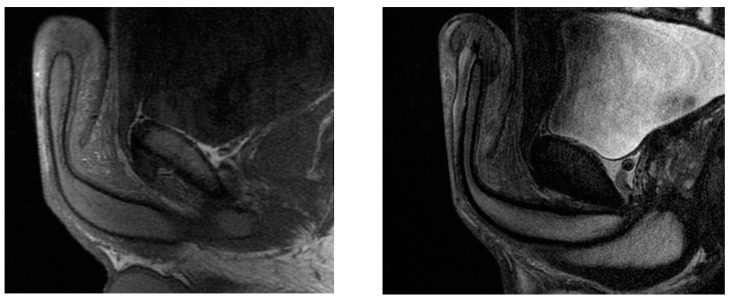
The paraffinoma was heterogeneous in T1 (on the **left**) and T2 (on the **right**).

**Figure 3 jcm-12-02604-f003:**
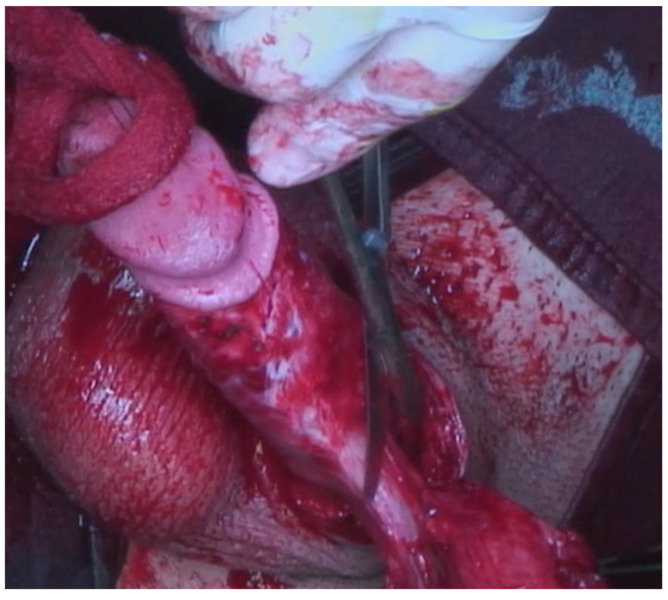
Excision of infiltrated tissues.

**Figure 4 jcm-12-02604-f004:**
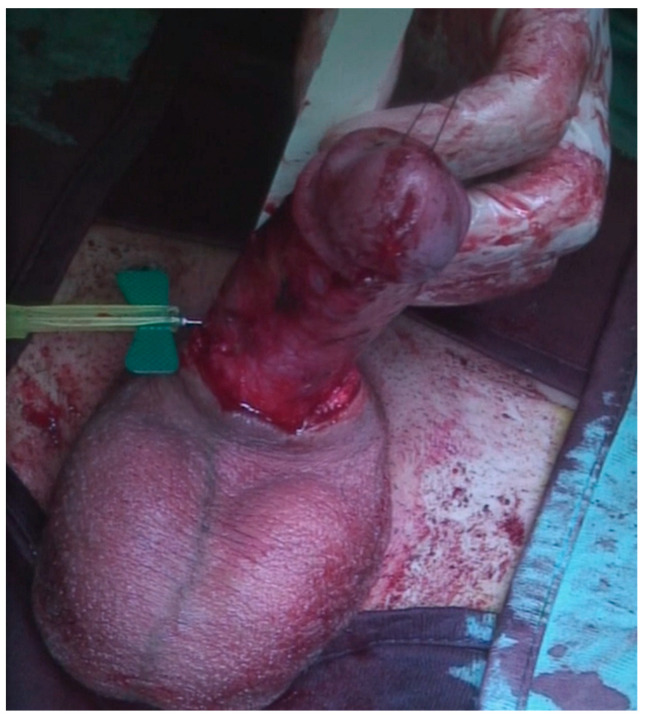
The defect was measured in length and girth.

**Figure 5 jcm-12-02604-f005:**
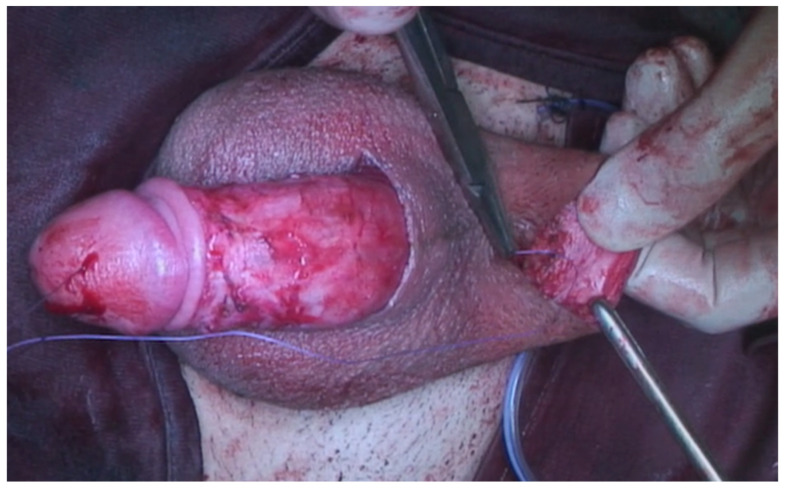
The subfascial scrotal tunnel was created and the penis was scrotalized.

**Figure 6 jcm-12-02604-f006:**
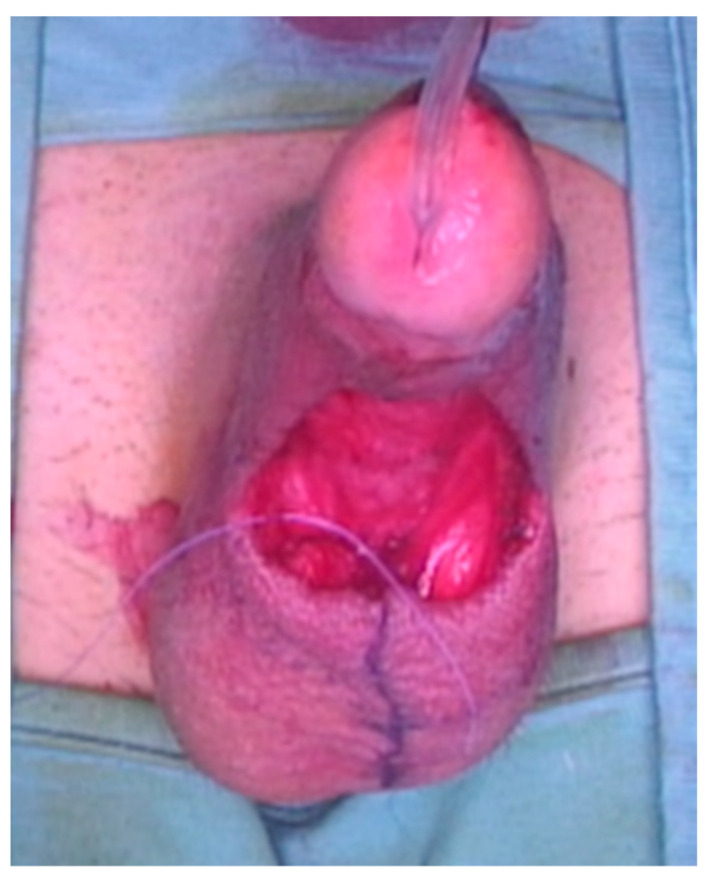
The scrotal access.

**Figure 7 jcm-12-02604-f007:**
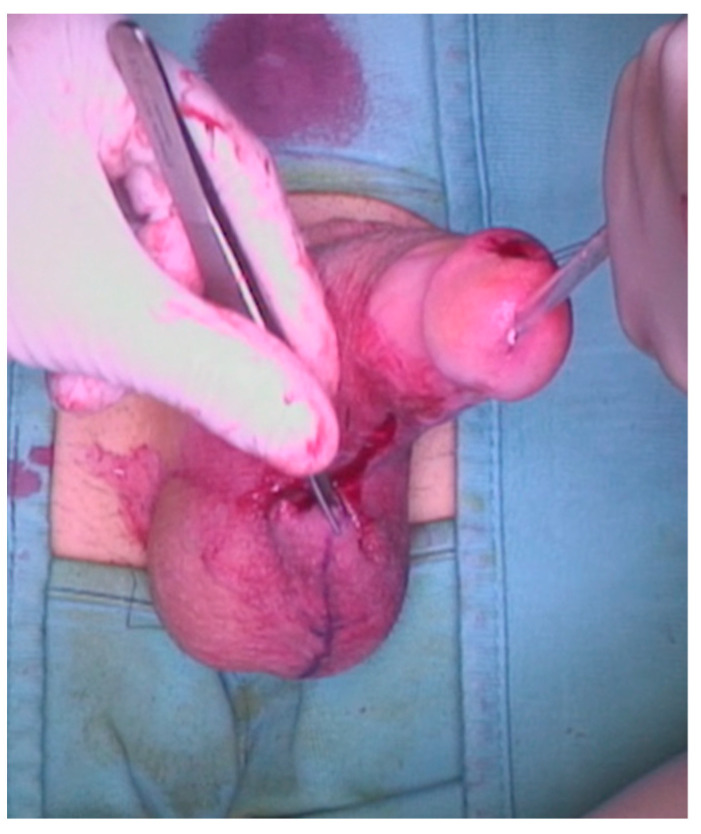
A V-Y scrotoplasty was performed.

**Figure 8 jcm-12-02604-f008:**
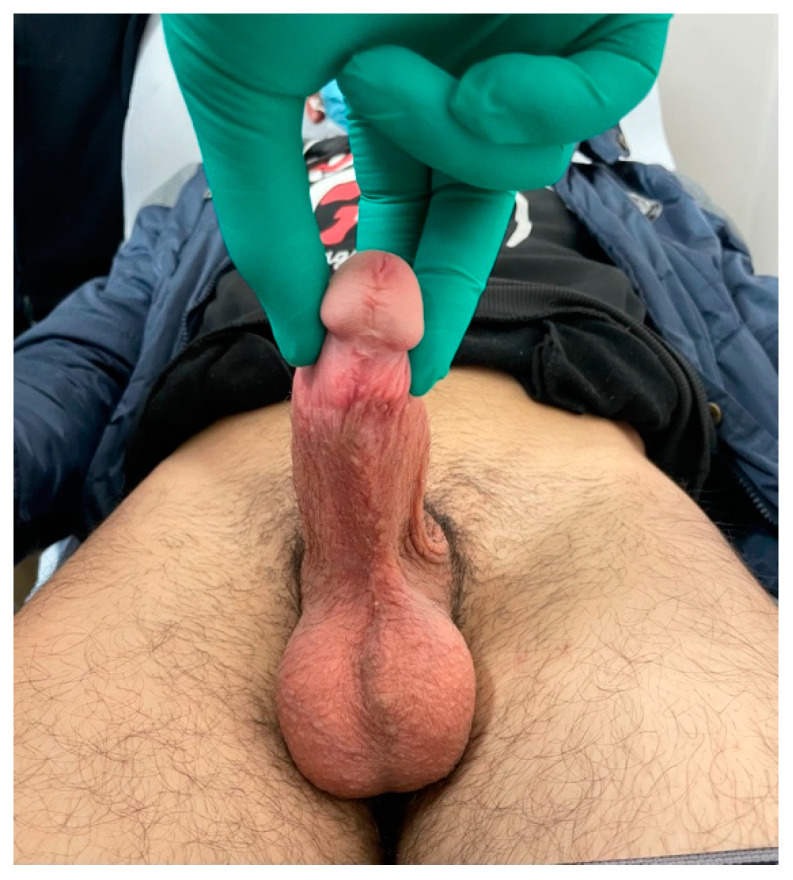
Final result at 6 month follow up.

**Figure 9 jcm-12-02604-f009:**
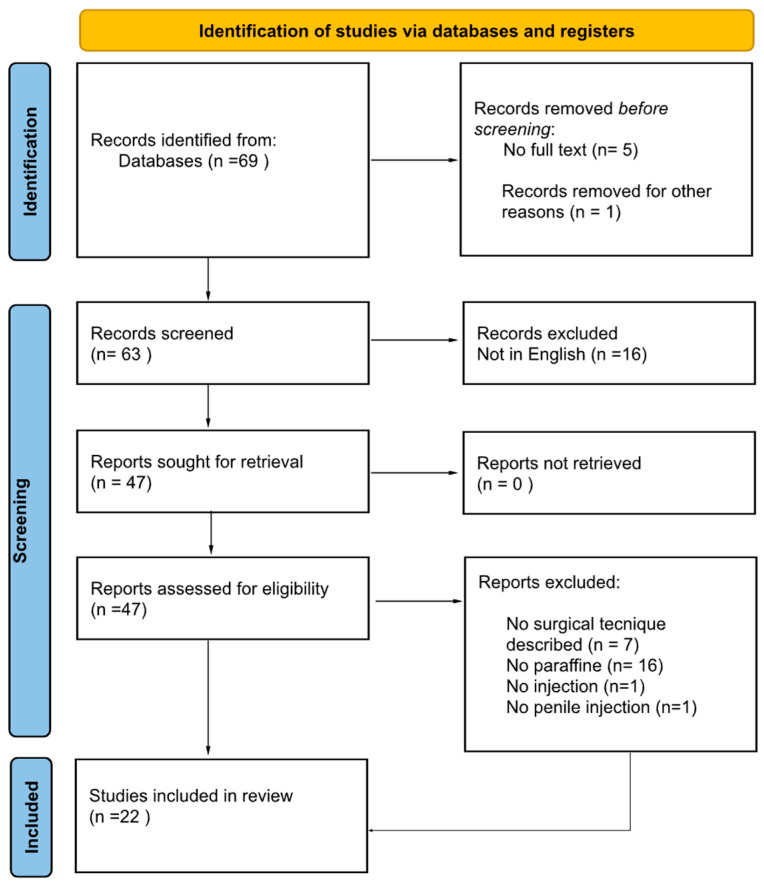
PRISMA statement.

**Figure 10 jcm-12-02604-f010:**
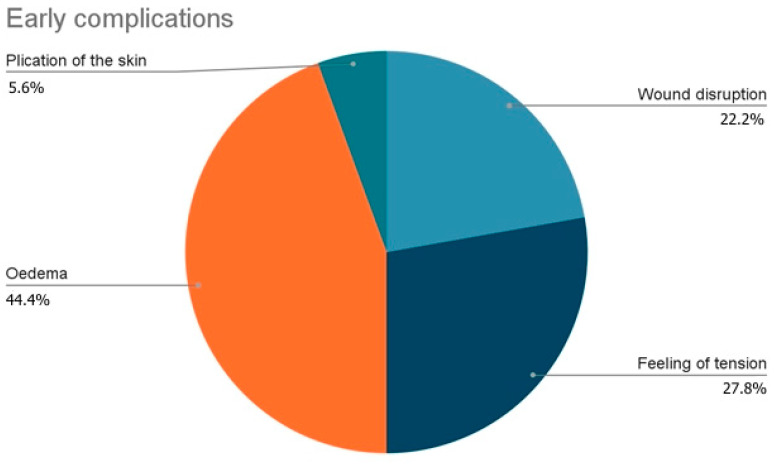
Distribution of early complications.

**Figure 11 jcm-12-02604-f011:**
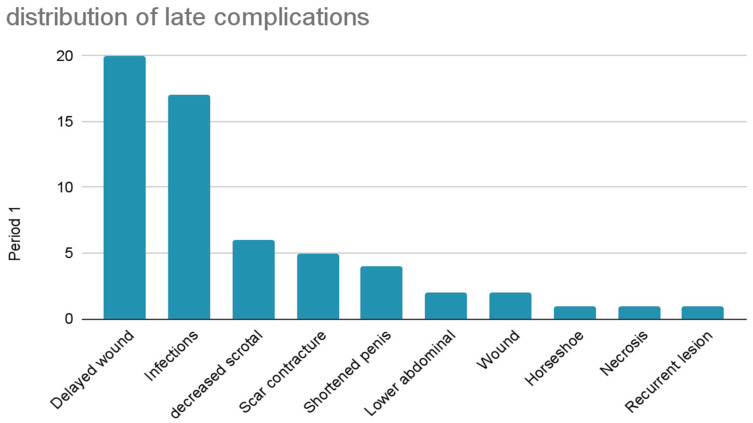
Distribution of late complications.

**Table 1 jcm-12-02604-t001:** Methodological quality of case reports and case series according to Murad et al. [16].

Domains	Selection	Ascertainment		Causality				Reporting
Leading Explanatory Questions	1	2	3	4	5	6	7	8
Track Lee [3]	yes	no	no	no	no	no	no	yes
Jeong [4]	yes	yes	yes	yes	no	no	yes	yes
Pehlivanov [17]	yes	yes	no	yes	no	no	no	yes
Katinka [18]	no	yes	yes	yes	no	no	no	yes
Picozzi [19]	no	yes	no	no	no	no	no	no
Bayraktar [20]	no	no	yes	yes	no	no	yes	no
Xeng Inn [12]	no	yes	no	yes	no	no	no	no
Seob Shin [21]	no	yes	yes	yes	no	no	yes	yes
De Siati [22]	no	yes	yes	yes	no	no	yes	yes
Cormio [22]	no	yes	yes	yes	no	no	yes	yes
Sun wook kim [23]	no	yes	yes	yes	no	no	yes	yes
Manjit Sigh [24]	no	yes	yes	yes	no	no	no	yes
Dellis [25]	no	yes	yes	no	no	no	yes	yes
Chon [26]	no	yes	yes	yes	no	no	yes	yes
Salaudin [27]	yes	yes	yes	yes	no	no	no	yes
Dunev Vladislav [28]	no	yes	yes	yes	no	no	no	yes
Jong Sung Kim [29]	no	no	yes	yes	no	no	no	yes
K.M. Danowsky [30]	no	no	yes	yes	no	no	no	yes
Vladislav [31]	no	yes	yes	yes	no	no	yes	yes
Boucher [32]	no	yes	yes	yes	no	no	no	yes
Ismy [33]	no	yes	yes	yes	no	no	yes	yes

Leading explanatory questions: 1. Does the patient(s) represent(s) the whole experience of the investigator (centre) or is the selection method unclear to the extent that other patients with similar presentation may not have been reported? 2. Was the exposure adequately ascertained? 3. Was the outcome adequately ascertained? 4. Were other alternative causes that may explain the observation ruled out? 5. Was there a challenge/rechallenge phenomenon? 6. Was there a dose–response effect? 7. Was follow-up long enough for outcomes to occur? 8. Is the case(s) described with sufficient details to allow other investigators to replicate the research or to allow practitioners make inferences related to their own practice?

**Table 2 jcm-12-02604-t002:** CARE checklist 2013.

	Title	Key Words	Abstract				Introduction	Patient Information				Clinical Findings	Timeline	Diagnostic Assessment				Therapautic Intervention			Follow-Up and Outcomes				Discussion				Patients Pepspective	Informed Conent
	1	2	3a	3b	3c	3d	4	5a	5b	5c	5d	6	7	8a	8b	8c	8d	9a	9b	9c	10a	10b	10c	10d	11a	11b	11c	11d	12	13
Thack Lee [3]	-	-	-	-	-	X	X	X	X	X	-	X	-	X	-	X	-	X	-	-	X	X	-	-	X	-	-	-	-	-
Jae Jo Jeong [4]	-	-	X		-	-	-	-	X	-	-	X	-	-	-	-	-	-	X	-	X	X	-	X	X	-	-	-	-	-
G Pehlivanov [17]	-	X	X	X	-	X	X	X	X	-	-	X	X	X	X	-	-	X	-	-	-	-	-	-	-	-	X	-	-	X
Katinka [18]	X	X	-	X	X	X	-	X	X	-	-	X	X	X	-	X	-	X	X	-	X	X	-	-	-	X	X	-	-	-
Picozzi [19]	-	-	-	-	-	-	-	X	X	X	-	X	-	X	-	X	-	X	-	-	-	-	-	-	-	-	-	X	-	-
Bayraktar [20]	-	-	X	-	-	-	X	X	X	-	-	X	-	X	-	X	-	X	-	-	X	X	-	-	X	X	-	X	-	-
Xeng Inn [12]	-	-	X	-	X	-	X	X	X	-	-	X	-	-	-	-	-	X	-	-	X	-	-	-	-	X	X	-	-	-
Seob Shin [21]	-	-	X	-	X	X	X	X	X	-	-	-	-	-	-	-	-	X	-	-	X	-	-	X	X	-	-	-	-	-
De Siati [34]	-	X	-	X	X	X	X	X	X	-	X	X	X	X	-	-	-	X	-	-	X		-	-	X	X	-	X	-	X
Cormio [22]	-	X	-	X	X	-	-	X	X	-	-	X	X	X	-	X	-	X	-	-	X	X	-	-	-	X	X	-	-	-
Sun wook kim [23]	-	X	-	X	X	-	X	-	X	-	-	-	X	-	-	-	-	X	-	-	X	-	-	-	X	X	X	X	-	-
Manjit Sigh [24]	-	X	-	-	X	-	X	X	X	-	-	X	-	X	-	-	-	X	-	-	X	X	-	-	-	-	-	X	-	-
Dellis [25]	-	-	X	X	X	X	X	X	X	-	X	X	-	-	-	X	-	X	-	-	X	X	-	-	X	X	X	X	-	-
Chon [26]	-	X	X	-	-	-	X	X	X	-	-	X	X	X	-	-	-	X	-	-	X	-	-	-	-	X	X	-	-	-
Salaudin [27]	-	X	X	-	X	X	X	-	X	-	-	X	-	-	-	-	-	X	-	-	-	X	-	-	-	-	X	X	-	-
Dunev Vladislav [28]	-	X	X	-	X	X	X	X	X	-	-	X	X	-	-	-	-	X	-	-	X	-	-	-	X	-	-	X	-	-
Jong Sung Kim [29]	-	X	X	X	X	X	X	X	X	-	-	-	-	-	-	-	-	X	-	-	X	X	X	X	X	X	X	X	-	-
K.M. Danowsky [35]	-	-	-	-	-	-	-	X	X	X	-	X	-	X	-	X	-	X	-	-	-	-	-	-	X	X	X	-	-	-
Dunev Vladislav [31]	-	X	X	X	-	-	X	X	X	-	X	X	X	-	-	-	-	X	-	-	X	X	-	X	-	X	-	X	-	-
Boucher [32]	-	X	X	-	X	X	X	X	X	-	X	-	-	X	-	-	-	-	-	-	X	X	-	X	X	X	X	X	-	-
Ismy [33]	-	-	X	-	X	X	X	X	X	-	-	X	-	-	-	-	-	X	X	-	X	X	X	X	-	-	X	X	-	X

1 The diagnosis or intervention of primary focus followed by the words“case report”; 2 2 to 5 key words that identify diagnoses or interventions in this case report, including “case report”; 3a Introduction: What is unique about this case and what does it add to the scientific literature?; 3b Main symptoms and/or important clinical findings; 3c The main diagnoses, therapeutic interventions, and outcomes; 3d Conclusion—What is the main “take-away” lesson(s)from this case?; 4 One or two paragraphs summarizing why this case is unique (may include references); 5a De-identified patient specific information; 5b Primary concerns and symptoms of the patient; 5c Medical, family, and psycho-social history including relevant genetic information; 5d Relevant past interventions with outcomes; 6 Describe significant physical examination (PE) and important clinical findings; 7 Historical and current information from the episode of care organized as a timeline; 8a Diagnostic testing (such as PE, laboratory testing, imaging, surveys) 8b Diagnostic challenges (such as access to testing, financial, or cultural); 8c Diagnosis (including other diagnoses considered) 8d Prognosis (such as staging in oncology) where applicable; 9a Types of therapeutic intervention (such as pharmacologic, surgical, preventive, self-care); 9b Administrationoftherapeuticintervention(suchasdosage, strength, duration); 9c Changes in therapeutic intervention(with rationale); 10a Clinician and patient-assessed outcomes(if available); 10b Important follow-up diagnostic and other test results; 10c Intervention adherence and tolerability(How was this assessed?); 10d Adverse and unanticipated events; 11a AscientificdiscussionofthestrengthsANDlimitationsassociatedwiththiscasereport; 11b Discussion of the relevant medical literature with references; 11c Thescientificrationaleforanyconclusions(includingassessmentofpossiblecauses); 11d The primary “take-away” lessons of this case report (without references) in a one paragraph conclusion; 12 The patient should share their perspective in one to two paragraphs on the treatment(s) they received; 13 Did the patient give informed consent? Please provide if requested.

**Table 3 jcm-12-02604-t003:** Classification of surgical complication according to Clavien Dindo.

Grade	Type of Complication
I	Oedema, penile rugae, scar contracture, decreased scrotal size
II	Delayed wound healing, wound disruption, wound deiscence, infection
IIIa	Necrosis, shortnered penis

## Data Availability

Not applicable.

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
