# Peer review of "Two-Stage Penile Reconstruction after Paraffin Injection: A Case Report and a Systematic Review of the Literature"

_jcm, 2023, doi:10.3390/jcm12072604_

Round 1

Reviewer 1 Report

This study reported two-stage penile reconstruction after paraffin injection, which is very rare now, and summarized the previously published literature about penile paraffin injection. However, the authors fail to present the pictures when the patient was finally healed, which is very important for the readers to learn about the final cosmetic result. Secondly, the authors summarized the previous similar literatures, yet it seems the references do not include most of cited papers, and I think the authors should list all the cited papers into references. Thirdly, since the paraffin injection for penile augmentation has long been not recommended for use, the references are almost 20 years ago, so the novelty is not adequate enough. 

Author Response

First of all, we wish to thank you for reviewing our paper. We realize that your time is valuable, and we are very grateful for this.

We thank the reviewer for her/his suggestion.

This study reported two-stage penile reconstruction after paraffin injection, which is very rare now, and summarized the previously published literature about penile paraffin injection. - However, the authors fail to present the pictures when the patient was finally healed, which is very important for the readers to learn about the final cosmetic result

We thank the reviewer for her/his suggestion. We added the figure 8, with the final result at 6 month follow-up.

-Secondly, the authors summarized the previous similar literatures, yet it seems the references do not include most cited papers, and I think the authors should list all the cited papers into references.

We thank the reviewer for her/his suggestion. We added the references 

-Thirdly, since the paraffin injection for penile augmentation has long been not recommended for use, the references are almost 20 years ago, so the novelty is not adequate enough.

 We thank the reviewer for her/his suggestion. We added the references

Reviewer 2 Report

The literature analysis is properly performed using the obtained data and the application of statistical tests.

The case presented which constitutes the pretext of this article is one of the easiest and does not seem to be difficult to solve even if other surgical procedures are used. It would be good to know if this procedure can be applied in all similar situations, regardless of the extent of the damages caused by the injection of substances (in the best case - paraffinoma).

Author Response

First of all, we wish to thank you for reviewing our paper. We realize that your time is valuable, and we are very grateful for this.

We thank the reviewer for her/his suggestion. The aim of the case report is to present an interesting reconstructive technique in the management of penile paraffinoma. Since the penile skin will be then removed such technique is also, please in patients with penile skin involvement and ulcerations. On the other hand, some patient features are mandatory (e.g. adequate scrotal size and no scrotal involvement) as reported in the article. Therefore, in all similar situations (adequate scrotal size, no scrotal involvement + penile skin involvement) the two stages surgical procedure is applicable. We do believe that the “easiest case” is a penile paraffinoma without skin involvement and that in these cases a paraffinoma excision with primary closure procedure could be worth, however we also believe that since these patients are complex in the clinical and surgical management and that related surgical procedures are often requiring different flaps or grafts harvesting and reconstructive surgical skills, no paraffinoma cases should be considered easy as the plan you have before surgery may change intra op. Regarding other surgical procedures that could be used, authors agree that simple excision of the paraffinoma with primary closure was not possible because the penile skin was infiltrated by paraffine, and that full-thickness or split-thickness skin grafting could not result in adequate penile appearance and patient aesthetic satisfaction. We believe that, if possible, a two-stage technique with scrotalization of the penis may improve aesthetic and functional outcomes compared with other flap-based surgical procedures (e.g., medial advancement flap technique and bilateral scrotal flap). The scrotal stretching obtained by erections during 6 months between the two stages and the delayed reconstructive procedure (scrotoplasty) allow a better vascular supply which leads to a more stretchable skin covering the penis. Of course, we know that more studies are needed to support such understandable theories, therefore more articles will be published by our institute comparing several different techniques in the management of such uncommon diseases

Reviewer 3 Report

   It was a very interesting study and within the scope of the journal. The author(s) had made a good attempt to share their experience in managing these cases and made good literature review. However, the following observations can be made:

1)      Does INTRODUCTION include: “What we know”, “What we don’t know” and “Aims”?

2)     Authors did not comment on the gap of knowledge the current research will try to fill.

3)     According to the general rule, the reference should be mentioned at the end of each sentence.

4)     Complications should be detailed according to Dindo Clavien system. 

5)     How missing data were addressed is not explained.

6)      As a general and unwritten rule, for every 1,000 to 1,500 words, a table or image is considered a standard article. Due to the fact that some data is expressed in the text, it is recommended to reduce the number of tables and images.

In results, there are described values that are already in the tables, making reading redundant. 

Limitations of the study were not presented.

9)      The conclusions should reflect upon the aims - whether they were achieved or not.

Author Response

First of all, we wish to thank you for reviewing our paper. We realize that your time is valuable, and we are very grateful for this.

We thank the reviewer for her/his suggestion.

It was a very interesting study and within the scope of the journal. The author(s) had made a good attempt to share their experience in managing these cases and made a good literature review. However, the following observations can be made:

1) Does INTRODUCTION include: “What we know”, “What we don’t know” and “Aims”? 

We thank the reviewer for her/his suggestion. ​​We modified the text accordingly.

2) Authors did not comment on the gap of knowledge the current research will try to fill.

 We thank the reviewer for her/his suggestion. ​​We modified the text accordingly.

3) According to the general rule, the reference should be mentioned at the end of each sentence.

We thank the reviewer for her/his suggestion. ​​We modified the text accordingly.

4) Complications should be detailed according to Dindo Clavien system.

We thank the reviewer for her/his suggestion. ​​We modified the text accordingly and we added a table

5) How missing data were addressed is not explained.

We thank you for her/his suggestion. Lukily there are no missing data.

6) As a general and unwritten rule, for every 1,000 to 1,500 words, a table or image is considered a standard article. Due to the fact that some data is expressed in the text, it is recommended to reduce the number of tables and images. In results, there are described values that are already in the tables, making reading redundant. 

We thank the reviewer for her/his suggestion. ​​We modified the text accordingly.

Limitations of the study were not presented.

 We thank the reviewer for her/his suggestion. We added limitations.

9) The conclusions should reflect upon the aims - whether they were achieved or not 

 We thank the reviewer for her/his suggestion. We modified the text accordingly.

Round 2

Reviewer 1 Report

This manuscript could be published after this revision.